# Dissolved organic carbon dynamics in a changing ocean: A COBALTv2 - ESM2M An ESM2M-COBALTv2 coupled model analysis

Lana Flanjak<sup>1,2</sup>, Aaron Wienkers<sup>3</sup>, and Charlotte Laufkötter<sup>1,2</sup>

and a 25% reduction in advection-dominated deep export at 1000 m depth.

**Correspondence:** Lana Flanjak (lana.flanjak@unibe.ch)

**Abstract.** Dissolved organic carbon (DOC) constitutes a major component of the marine carbon cycle, yet its present contributions to carbon export, and the response to future climate change remain poorly constrained. Using COBALTv2-ESM2M-COBALTv2 — GFDL's ocean biogeochemistry model COBALTv2 coupled to the ESM2M Earth System Model — we evaluate present-day DOC distribution and export and project their responses to a high-emission future scenario RCP8.5 to the year 2100.

Our model reproduces well the observed large-scale DOC patterns, with highest concentrations ( 70– $80~\mu$ mol Ckg $^{-1}$ ) in subtropical gyres and lower values ( 40– $50~\mu$ mol Ckg $^{-1}$ ) in subpolar and equatorial upwelling regions. Biological DOC production and remineralization rates are highest in nutrient-rich upwelling zones. The net DOC produced is then transported to the stratified oligotrophic gyres where DOC accumulates, thereby forming the observed global DOC distribution. Present-day global DOC export at 100~m is estimated at  $1.6~pgC~yr^{-1}$ , accounting for about 25~19% of the total organic carbon (TOC) export modeled at that depth. By 1000~m, DOC export decreases sharply to  $0.09~pgC~yr^{-1}$ , solely because microbial remineralization removes a significant fraction of DOC as it descends deeper into the water column. At 100~m, globally integrated mixing-mediated export is nearly twice that of advection, especially in boundary current regions and subpolar gyres where strong seasonal mixing occurs, whereas advection dominates in subtropical gyres via large-scale subduction of accumulated DOC. At 1000~m, however, advection dominates, particularly in the North Atlantic where deep-water formation facilitates DOC export. Under future warming, intensified stratification and reduced nutrient supply drive a net decline in global DOC production. Nevertheless, upper-ocean DOC concentrations increase slightly, underscoring the continued importance of physical transport in redistributing DOC. The model projects a 6% reduction in DOC export at 100~m, driven primarily by weakened mixing,

## 1 Introduction

Dissolved organic carbon (DOC) constitutes one of the largest pools of bioreactive carbon in the ocean. Its vast inventory is approximated at 662 PgC, comparable to the quantity of inorganic carbon in the atmosphere (Carlson and Hansell, 2015; Hansell et al., 2009). DOC plays an important role in the biogeochemical carbon cycling, and while the majority of modeling

<sup>&</sup>lt;sup>1</sup>Climate and Environmental Physics, Physics Institute, University of Bern, Bern, Switzerland

<sup>&</sup>lt;sup>2</sup>Oeschger Center for Climate Change Research, University of Bern, Bern, Switzerland

<sup>&</sup>lt;sup>3</sup>Environmental Physics, Institute of Biogeochemistry and Pollutant Dynamics, ETH Zürich, Zürich, Switzerland

studies has focused on particulate organic carbon (POC) in the context of export production, DOC is estimated to contribute 20% to total organic carbon (TOC) export (Hansell et al., 2009). However, this estimate comes with considerable uncertainties given the vast range in contemporary POC export estimates (Henson et al., 2022; Laufkötter et al., 2016; Bopp et al., 2013). Compounding this, the mechanisms behind DOC cycling and export - and their response to climate change - remain poorly constrained.

The predominant source of DOC is the extracellular release of photosynthate and other metabolites by phytoplankton (Carlson and Hansell, 2015). Complementary biochemical mechanisms affecting the DOC pool include (i) grazing and excretion by zooplankton and higher predators (Moran et al., 2022; Strom et al., 1997); (ii) release of cellular material via cell lysis by bacteria and vira (Lønborg et al., 2013; Keller and Hood, 2011; Middelboe and Lyck, 2002); and (iii) solubilization of POC to DOC by bacteria (Jiao et al., 2010; Nagata et al., 2000). The primary sink of DOC occurs through bacterial remineralization into dissolved inorganic carbon. These processes are profoundly influenced by ocean warming, both directly, through its impact on physiology and metabolic rates, and indirectly, through enhanced water-column stratification and ensuing decreased nutrient supply (Sallée et al., 2021; Bopp et al., 2013). Several studies have demonstrated that warming and elevated CO<sub>2</sub> levels enhance primary production, which, in turn, increases the release of DOC. This shift has implications for trophic structure and carbon export, as POC and DOC have distinct roles in the ecosystem and are exported by different mechanisms. However, reduced nutrient availability, driven by increased stratification, is expected to decrease overall primary production and consequently limit the production of both DOC and POC. Additionally, as respiration rates rise with warming, the rapid utilization of DOC by microbial consumers—often referred to as the strengthening of the microbial loop—may counteract DOC accumulation (Lønborg et al., 2020).

The likelihood and timing of DOC being either remineralized or accumulated and exported are governed by its lability. DOC lability largely depends on its chemical composition, as well as environmental conditions, including abiotic factors and ecosystem composition (e.g., prokaryotic diversity) (Lønborg et al., 2020; Wagner et al., 2020). As a result, DOC is conventionally divided into different pools based on turnover times: labile (hours to days), semi-labile (weeks to months), semi-refractory (years to decades), and refractory (millennia). Labile DOC is very reactive and it serves as an easily available substrate for heterotrophic bacterial production, thereby not playing a significant role in carbon export. Semi-labile, semi-refractory, and refractory DOC, because of their slower turnover times, can accumulate and are susceptible to export to the deeper ocean.

Identifying the processes driving DOC dynamics and understanding their responses to future climate change holds relevance for improving constraints of the marine carbon cycle. Global ocean biogeochemical models are potent tools enabling analyses of carbon cycle across a range of spatial and temporal scales, including its responses to past and future climate changes. However, DOC export and dynamics, and their responses to future climate change, have been sparsely investigated in model studies. Available studies have primarily estimated DOC export under contemporary ocean conditions (Nowicki et al., 2022; Roshan and DeVries, 2017; Hansell et al., 2009), with some considerations of how the expansion of subtropical gyres - where DOC contributes to up to half of TOC export - might influence overall carbon export dynamics (Roshan and DeVries, 2017).

The relative roles of advection and mixing in driving DOC export, as well as their potential future changes, remain largely unexplored in an Earth System Model (ESM).

The only existing estimate of a potential future reduction in DOC export, reported by Sreeush et al. (2024), projects a 4% decrease below 100 m by 2100 under SSP3-7.0 scenario. However, it lacks an analysis of the underlying physical and biogeochemical drivers or a regional breakdown. To our knowledge, no studies have explicitly and comprehensively analyzed responses of DOC export, including its drivers and mechanisms, within a future climate projection using a fully coupled ESM. This leaves a significant gap in our understanding of the role of DOC in carbon cycling under climate change. By addressing this gap, we enhance constraints and advance our knowledge of the marine carbon cycle in a warming climate.

## 65 2 Methods

70

In this study, we examine DOC dynamics and export under present and future climate conditions in reference to the high-emission scenario RCP8.5. To this end, we analyze simulation output from the Carbon, Ocean Biogeochemistry and Lower Trophics version 2 (COBALTv2) model (Stock et al., 2020) embedded into the Modular Ocean Model version 5 (MOM5) of the fully coupled Geophysical Fluid Dynamics Laboratory Earth System Model version 2M (GFDL ESM2M) (Dunne et al., 2012).

# 2.1 Physical model configuration

The GFDL ESM2M is a fully coupled ESM comprising integrated components representing the atmosphere, ocean, land, and sea ice, coupled through the Flexible Modeling System (FMS) infrastructure to conserve energy, mass, and tracer fluxes across component boundaries.

The physical ocean component (MOM5) utilises a tripolar horizontal grid with a nominal resolution of 1° increasing zonally to 1/3° near the equator. The vertical discretisation consists of 50 levels with thickness increasing from 5 m at the surface to 170 m in the abyssal ocean. The model uses a rescaled vertical coordinate (z\*) which adjusts cell thickness with variations in sea surface height. MOM5 simulates ocean circulation through resolved dynamics and parametrised subgrid-scale processes, including mesoscale eddy-induced advection (Gent-McWilliams scheme), neutral diffusion, and vertical mixing processes.

Vertical mixing is represented through a constant background diffusivity augmented by a tidal mixing parametrisation and the K-profile parametrisation (KPP) for boundary layer mixing. The model resolves the advection of physical and biogeochemical tracers using the MDPPM numerical scheme.

The atmospheric component consists of the Atmospheric Model version 2 (AM2) with a horizontal resolution of  $2^{\circ} \times 2.5^{\circ}$  and 24 vertical pressure levels. The sea ice component (SIS) shares the same horizontal grid as the ocean component and employs a three-layer thermodynamic model for sea ice. The land component implements the Land Model version 3.0 (LM3.0), which simulates water, energy, and carbon cycling between vegetation and soil.

## 2.2 Biogeochemical model configuration

## 2.2.1 Ecosystem dynamics

The COBALTv2 model uses 33-tracers to simulate global scale dynamics of carbon, nitrogen, phosphorus, iron, and oxygen. It models the dynamics of a plankton food web organized by size and functional types into three classes of phytoplankton, three classes of zooplankton and a group of heterotrophic free-living bacteria. Phytoplankton types are represented as small, large, and diazotrophic phytoplankton. The distinction between small and large phytoplankton corresponds to an equivalent spherical diameter of about 10  $\mu$ m, while diazotrophs are parameterized as Trichodesmium cyanobacteria. Phytoplankton take up nutrients only in inorganic form, (NO<sub>3</sub>), ammonium (NH<sub>4</sub>), phosphate (PO<sub>4</sub>), dissolved inorganic iron (Fe) and silicate (SiO<sub>4</sub>). The uptake of NO<sub>3</sub> and NH<sub>4</sub> is modeled after O'Neill et al. (1989) where the presence of NH<sub>4</sub> inhibits NO<sub>3</sub> uptake and similarly, the presence of NO<sub>3</sub> inhibits NH<sub>4</sub> uptake. Diazotrophs combine N<sub>2</sub>-fixation and, if available, dissolved inorganic nitrogen to meet their nitrogen requirement (Holl and Montoya, 2005). Silica is only taken up by large phytoplankton. The uptake of nutrients is modeled after Michalis-Menten kinetics, assigning significantly higher lower half-saturation constants to small phytoplankton to represent the benefits of higher surface area to volume ratio in nutrient uptake. Nutrient and light co-limit photosynthesis, based on the formulation from Geider et al. (1997). Small phytoplankton exhibit greater chlorophyllspecific light absorption rates compared to large phytoplankton, attributed to their higher surface area to volume ratio. Nutrient limitation is modeled after Liebig's Law of the Minimum (von Liebig, 1840). The elemental ratios of phytoplankton are fixed but slightly diverge from Redfield ratio: small phytoplankton exhibit an N:P ratio of 20:1, diazotrophs 40:1, and large phytoplankton 12:1. Zooplankton are represented by three size-based classes: small, medium, and large zooplankton. Small zooplankton graze on bacteria and small phytoplankton, medium zooplankton graze on diazotrophs, large phytoplankton, and small zooplankton, and large zooplankton graze on diazotrophs, large phytoplankton, and medium zooplankton. Medium and large zooplankton are consumed by an implicit higher trophic-level predator. Zooplankton biomass-specific grazing rates decrease as their size increases (Hansen et al., 1994). Grazing follows a Holling's Type II functional response, incorporating density-dependent prey selection. Grazing half-saturation constants are the same for all of the zooplankton classes but calibrated against observed global patterns of phytoplankton biomass. Bacteria take up labile dissolved organic carbon. They are consumed by small zooplankton and subjected to implicit density-dependent mortality due to vira. Temperature-sensitive biological processes, namely phytoplankton nutrient uptake and growth, zooplankton grazing, bacterial uptake of DOC, are modeled after Eppley (1971) formulation:  $T = e^{kT}$ , where T is the temperature in °C, and k temp is the temperature scaling factor, set to  $0.063\,^{\circ}\text{C}^{-1}$ . This value corresponds to a Q10 of 1.88, indicating that reaction rates nearly double for every 10  $^{\circ}\text{C}$  increase in temperature. Detailed formulations of the COBALTv2 ecosystem equations are provided The development of the model was carried out and described in detail, together with additional ecosystem equations, by Stock et al. (2020).

## 2.2.2 DOC fluxes




Although nitrogen is the central currency of the model, the fixed C:N:P ratio within the model allows us to translate the dissolved organic matter results into carbon terms, i.e., DOC. The main fluxes affecting DOC pools are quantified as present-

day (1990-2010) averages and illustrated in Fig. 1. Hereon, future averages and future changes refer to the 2080-2000 period. DOC is categorized into three pools based on its lability; semi-refractory, semi-labile, and labile DOC. Labile DOC is the fastreacting pool, taken up by bacteria and remineralized into inorganic carbon and nutrients. This pool is also the only temperaturesensitive DOC pool, as its remineralization depends on bacterial metabolism, which is modeled to have a temperature dependency (as described in the previous paragraph). Semi-labile DOC decays over seasonal timescales, while semi-refractory DOC decays over annual to decadal timescales, both ultimately decaying into the labile DOC pool. The refractory component is represented by a background of 42  $\mu$ mol of DOC. Due to its millennial turnover timescales, its dynamics are not expected to significantly influence the shorter timescales analyzed in this study. Net primary production (NPP) is the primary source of organic matter in the system. The fraction of NPP exuded as labile DOC is constant at 13%. Zooplankton egest 30% of what they graze on, while the rest is assimilated. Egested organic matter is partitioned between particulate (detritus) and dissolved organic matter, depending on the zooplankton size class. Partitioning of egested organic matter to DOC is modeled to be 1/6 for small zooplankton, 2/3 for medium zooplankton, where the dissolved matter is then distributed among labile (70%), semi-labile (20%) and semi-refractory (10%) DOC pools. Egestion of large zooplankton results in particulate detritus only. Small zooplankton and bacteria are subjected to constant losses due to viruses (viral infections), both resulting in 20% of their respective production directed to DOC pools. For small zooplankton, the fractionation between labile, semi-labile, and semi-refractory DOC is 70%, 20%, and 10%, while for bacteria it is 65%, 20%, and 15%, respectively. Semi-labile and semirefractory DOC decays into labile DOC by rates of  $0.01 \,\mathrm{day}^{-1}$  and  $0.1 \,\mathrm{yr}^{-1}$ , according to their typical turnover timescales. The pool of highest reactivity, labile DOM, is taken up and remineralized by bacteria. The riverine input contributes a combined 0.22 PgC yr<sup>-1</sup> of dissolved inorganic and organic carbon, with the latter comprising labile, semi-labile, and semi-refractory DOC (Stock et al., 2020).






To quantify the physical transport mechanisms of DOC, we extracted the vertical advective and turbulent mixing fluxes directly from the MOM5 ocean model component. The advective flux components were computed as the product of the three-dimensional velocity field and DOC concentration. Turbulent mixing fluxes were derived from the internal physical mixing model, which includes vertical diffusion, K-profile parameterisation, and additional mixing processes (Dunne et al., 2012). These flux divergence terms are finally integrated to obtain total DOC transport through specified depth horizons (100 m and 1000 m), enabling the separation of advection and mixing contributions to DOC export. To quantify the physical transport mechanisms of DOC, we extracted the vertical advective and turbulent mixing fluxes directly from the MOM5 ocean model component using a Finite Volume Method. For advective fluxes, the volume flux across each cell face is multiplied by the DOC concentration of the donor cell, capturing both horizontal and vertical transport. Turbulent mixing fluxes are derived from the internal physical mixing model, which includes vertical diffusion, K-profile parameterisation, and additional subgrid-scale processes (Dunne et al., 2012). DOC export at specified depth horizons (100 m and 1000 m) is calculated by identifying the cell edge closest to the target depth and accumulating the exact flux through that level as a diagnostic at the model's output frequency. This method captures both downward and upward DOC transport; negative flux values indicate net upward movement (e.g., due to upwelling or turbulent mixing). The Finite Volume implementation ensures exact budget closure to

**Figure 1.** Globally integrated DOC fluxes and grazing fluxes between food-web components in COBALTv2. Fluxes represent annual averages for the present-day period (1990–2010), integrated over the upper 100 m of the water column. Fluxes implicitly represented as viral losses are denoted by the term *vir\_loss*. Semi-labile (SLDOC) and semi-refractory (SRDOC) DOC decay fluxes were calculated by multiplying their mean pool sizes by their respective first-order decay rates. DOC export fluxes at 100 m and 1000 m via corresponding pools are also shown. All values are expressed in Pg C yr<sup>-1</sup>.

machine precision, thereby enabling a precise separation of advective and mixing contributions to DOC export within each fixed-depth control volume.

#### **2.2.3** Tuning


COBALTv2 was designed and tuned within the GFDL ESM4.1 Stock et al. (2020), driven by the MOM6 physical ocean model. To account for the resulting physical model differences compared to MOM5 within ESM2M (e.g. resolution and grid diffusivity and the follow-on influence on the large-scale physical circulation and transport fields), the biogeochemical model parameterisations were re-calibrated following the methodology established in Stock et al. (2020). Key biogeochemical free parameters were adjusted to achieve agreement with observational constraints including global primary production, ecosystem community composition, nutrient distribution/limitation, and carbon and nitrogen system variables. We summarize the key

tuning differences in Table A1 in the Appendix, which lists the parameters that were adjusted and their values across model versions.

# 165 2.3 Simulation protocol

The simulation protocol followed the CMIP5 experimental design with modifications specific to our biogeochemical analysis. Each simulation was initialised from a 500-year pre-industrial spin-up of the updated COBALTv2 implementation, which consequently began from the equilibrated 1000-year pre-industrial spin-up run at GFDL as part of the CMIP5 submission. This statistical equilibrium of the coupled physical-biogeochemical system was evidenced by the globally integrated DOC inventory trends below 0.01% per decade.

From this pre-industrial statistical equilibrium, we subsequently conducted simulations for three periods: (1) a land use spin-up (1700–1860) with preindustrial radiative forcing but time-varying land use changes; (2) a historical period (1860–2010) with observed atmospheric greenhouse gas concentrations, aerosol forcing, solar variability, and volcanic eruptions; and (3) a future projection phase (2010–2100) following the RCP8.5 high-emission scenario. A pre-industrial control simulation from the same initial conditions, integrating from 1700–2100 was also run alongside these experiments for comparison.

#### 3 Results & Discussion



# 3.1 Present-day DOC distribution and comparison with observations

The modeled present-day DOC distribution in the upper 100 m of the ocean and corresponding observational data are presented in Fig.2a and c. Highest DOC concentrations of 70-80  $\mu$ mol Ckg<sup>-1</sup> are simulated in subtropical and tropical ocean regions, with lower concentrations of 40-50  $\mu$ mol Ckg<sup>-1</sup> in subpolar regions and parts of equatorial upwelling zones. The simulated distribution shows broad consistency with observed values and patterns, as evidenced by a spatial correlation of 0.73, a small bias of -1.08  $\mu$ mol Ckg<sup>-1</sup>, and an rmse of 20.88  $\mu$ mol Ckg<sup>-1</sup>. At 1000 m depth, the model simulates low DOC concentrations of 40  $\mu$ mol Ckg<sup>-1</sup> across most of the ocean, with higher concentrations of up to 50  $\mu$ mol Ckg<sup>-1</sup> in the North Atlantic (Fig. 2c,d).

A comparison with other state-of-the-art biogeochemistry models indicates that the DOC distribution simulated by ESM2M-COBALTv2 is comparable to or outperforms that of other models. A thorough evaluation of DOC patterns by Stock et al. (2020) in the original configuration of COBALTv2, coupled to the CMIP6-generation model ESM4.1, similarly reveals strong agreement between modeled and observed DOC data. Comparable agreements are reported using other biogeochemical models, such as CESM2, where Sreeush et al. (2024) demonstrate a reasonable fit with observations. Lennartz et al. (2024), using a modified ESM, achieved a spatial correlation of approximately 0.6 by incorporating a tuned temperature dependency of bacterial growth efficiency and macronutrient co-limitation on DOC uptake, significantly improving the model's ability to reproduce observed patterns incorporating macronutrient co-limitation on DOC uptake, thereby achieving a spatial correlation

of  $R^2 = 0.55$  for the surface ocean and  $R^2 = 0.75$  when integrated over depth. Our model integrates temperature and multinutrient dependencies, providing an additional mechanistic layer to capture DOC cycling dynamics.



An area with significant disagreement between observations and model data is the Arctic, where observations show much higher values. This discrepancy can be attributed to several factors. Riverine inputs play a crucial role in Arctic DOC dynamics, with highly seasonal variability. More than 90% of the annual river discharge into the Arctic Ocean occurs between May and June, with DOC concentrations increasing in tandem with increased water flow, leading to a peak in early summer (Fouest et al., 2013; Dittmar and Kattner, 2003; Cauwet and Sidorov, 1996). Observations are typically collected during this ice-free season, resulting in a temporal bias toward capturing the highest DOC concentrations. In contrast, the model (COBALTv2-ESM2M) (ESM2M-COBALTv2) does not dynamically resolve the riverine DOC contributions and their seasonality, leading to the apparent offset between the modeled and observed values but model uses prescribed, climatological concentrations for river carbonate constituents, thus only resolves the temporal variability of river DOC contributions due to freshwater variability but not due to DOC concentrations.

**Figure 2.** Comparison of present-day modeled average DOC concentrations with observational data. Left panel figures a) and c) are based on model output and right panel figures (b and d) are based on observations compiled in Hansell et al. (2021). Top panel figures (a and b) show DOC concentrations averaged over top 100 m, while in the bottom panel (c and d) they are averaged over depth range of 990 - 1100 m. We express DOC concentration values in units of  $\mu$ mol Ckg<sup>-1</sup> for comparability with other studies.

The present-day distribution of DOC in the ocean is a result of different drivers at play, including the main biological controls of DOC concentration and content, as well as the physical transport of DOC. DOC is largely generated in the high productivity zones of the surface ocean such as equatorial upwelling region and in the high latitudes, e.g., along western boundary currents, from where it is largely remineralized as well, and the net DOC is transported by ocean currents to subtropical gyres where it accumulates reaching the described high concentrations (Najjar et al., 2007). The subtropical ocean is highly stratified and the input of nutrients, including upward mixing and upwelling, is very limited leading to low productivity, both in terms of primary production and bacterial production. The distribution at 1000 m depth reveals low concentrations in vast ocean areas, with contrasting high concentration occurring in North Atlantic where the DOC-rich subtropical waters are transported via deep convection into the deep-water formation zone.

Figure 3a shows latitudinal transects of the Atlantic, Pacific, and Indian Oceans, based on model data, highlighting the accumulation of DOC in the centers of subtropical gyres across the three major ocean basins. DOC concentrations reach average values of about 65-70 μmol Ckg<sup>-1</sup> in the gyre centers. Concurrently, isopycnal surfaces deepen in the gyre centers, indicating waters with higher DOC concentration propagating into depth. This can be explained by the convergence of seawater in the centers of the gyres, where at the surface DOC accumulates and is downwelled further to depth due to negative Ekman pumping (Levy et al., 2013). In these areas, isopycnal surfaces indicating DOC values of ≥ 45 μmol Ckg<sup>-1</sup> extend towards mesopelgaic depths of 400 − 500 m. A deep-water formation zone feature is observed around 60°N in the Atlantic, where a localized area of higher DOC concentration extends deeper than surrounding regions.

**Figure 3.** Distribution of modeled DOC concentrations across depth in ocean basins, presented through latitudinal transects (a) and evaluated through model-observation comparisons (b). The latitudinal transects are based on present-day (1990–2010) model output, with isopycnal lines indicating water masses with similar DOC concentrations. The scatter plot compares modeled and observed DOC concentrations, using present-day (1990–2010) model output and observations from Hansell et al. (2021). A 1:1 reference line (red dashed) and regression statistics (R<sup>2</sup> and rmse) are included to evaluate model performance and and validate the accuracy of the latitudinal transects in the left panel.

# 3.2 Biological sources and sinks as primary constraints on DOC concentration and distribution




DOC source processes that are hereafter called DOC production processes include phytoplankton exudation (~40% of the total DOC production), zooplankton egestion (~10% of total DOC production), and viral losses (~50% of total DOC production). These relative contributions to DOC production hardly change over the simulation period. Highest DOC production rates take place in the equatorial upwelling zones where well-mixed waters bring a lot of nutrients enabling high productivity of the system (Fig. 3a). Due to similar mechanisms, high DOC production rates also arise along western boundary currents and in some coastal areas. The lowest DOC production zones are found in the stratified oligotrophic subtropical gyres and high-nutrient-low-chlorophyll zones, such as vast parts of Southern Ocean. The only modeled DOC sink is remineralization by heterotrophic bacteria. As bacterial communities graze on DOC and are also co-limited by nutrient availability, remineralization rates exhibit a similar spatial distribution to DOC production (Fig. 3e) (Fig. 4c). However, production and remineralization processes are not tightly coupled due to variations in the lability of DOC, implicitly capturing biochemical and physicochemical factors that prevent bacterial communities from fully depleting DOC. While labile DOC is largely consumed by bacteria in the upper ocean, semi-labile and semi-refractory DOC contribute to longer-term DOC accumulation. This decoupling results in areas of net positive DOC production. The spatial distribution of net DOC production is illustrated in Fig. 3e Fig. 4e. This pattern closely aligns with biologically productive regions (Fig. 4a,c). The equatorial Pacific region stands out as

a hotspot of DOC production, exceeding 1.5 kgC m<sup>-2</sup> yr<sup>-1</sup>. Other productive areas of the ocean are equatorial upwelling regions in general, and temperate ocean that exhibit a highly seasonal cycle in productivity. Ocean currents then distribute this accumulated DOC, shaping the global distribution pattern observed in Fig. 2. We further explore and focus on the role of ocean circulation in shaping large-scale DOC dynamics in the Sect. 3.4.






Our DOC production pattern mirrors that of another ESM (Sreeush et al., 2024) and aligns closely with the pattern described by Hansell and Carlson (2001), but differs significantly from a DOC restoring approach by Roshan and DeVries (2017). The authors restore DOC in a circulation model to a neural network-derived global DOC estimate, bypassing biogeochemical feedbacks. In their model, net DOC production is high in the subtropical gyres and low in the equatorial upwelling area. These high subtropical net DOC production rates (reaching  $\sim$ 15 gC m<sup>-2</sup> yr<sup>-1</sup>) are surprising given the strong nutrient scarcity that limits biological production in these regions. Our modeled net DOC production average for subtropical gyres ( $\sim$ 2 gC m<sup>-2</sup> yr<sup>-1</sup>) appears to align more closely with established biogeochemical expectations.

On the one hand, the fast surface currents of the equatorial Pacific require small time steps to be able to distinguish between production within the gyres and transport into the gyres. The 1-year timestep used by the transport matrix in the Roshan and DeVries (2017) study may falsely assign fast transport of DOC into the gyres to DOC production within the gyres.

On the other hand, some of the assumptions underlying DOC dynamics in COBALTv2 may require further refinement. To replicate their pattern in a fully-coupled marine biogeochemistry model, one would need to allow small pico/nanoplankton in the subtropical gyres to produce longer-lived semi-labile or semi-refractory DOC, and require larger plankton in upwelling areas to produce very little or short-lived DOC.

The projected future change in DOC production and remineralization (Fig. 3b,d) reveal distinct regional responses to environmental changes. North Atlantic shows a pronounced decrease in both DOC production and remineralization (blue regions, approximately -0.3 to -0.5 kgC m<sup>-2</sup> yr<sup>-1</sup>). The equatorial Pacific displays a band of decreased biological DOC cycling (encompassing both production and bacterial remineralization), which could be related to changes in upwelling intensity or nutrient availability. A notable hotspot of increased DOC cycling (red regions, +0.3 to +0.5 kgC m<sup>-2</sup> yr<sup>-1</sup>) appears in the South Atlantic gyre along the Benguela current, indicating intensification of biological activity in this region. The Southern Ocean exhibits a mixed response, with large areas showing decreased rates of biological DOC cycling.

Globally integrated DOC production rates show interannual variability but exhibit an overall steady decline of -2.4 TgC yr<sup>-1</sup>. This decrease is likely driven by warming-induced stratification and reduced nutrient supply (Appendix, Fig. A2), which are key factors influencing biological DOC production. Remineralization rates, meanwhile, decline at a slower rate of -1.3 TgC yr<sup>-1</sup>, possibly reflecting a mismatch between bacterial activity and primary production responses to rising upper ocean temperatures and associated physicochemical changes. Large areas of the ocean show significant mixed-layer shoaling, particularly pronounced in the Southern Ocean and North Atlantic where changes exceed 40 m (Appendix, Fig. A2). This widespread shoaling of the mixed layer could restrict the extent of vertical mixing and nutrient supply to the surface ocean, potentially explaining the projected decline in DOC production rates (-2.4 TgC yr<sup>-1</sup>). However, some regions like parts of the equatorial Pacific show mixed-layer deepening, suggesting regional variations in the factors controlling upper ocean structure.

These projected trends highlight the interplay between physical and biogeochemical drivers in shaping DOC dynamics under a future climate scenario. In interpreting these results, it is important to consider how certain model assumptions may influence the robustness of future projections. Our modeled distribution of DOC and other relevant biogeochemical variables aligns well with present-day observations (Fig. 2; Appendix, Fig. A1), although temperature-sensitive remineralization is currently applied only to labile DOC. Semi-labile and semi-refractory DOC pools use empirically based constant decay rates reflecting current conditions, so future temperature effects are partially captured through remineralization into labile DOC. While suitable for analysis of present-day cycling, this approach may introduce some uncertainty in future projections. Ongoing model development aims to include environmentally sensitive decay rates for these pools to improve projection robustness. Similarly, DOC production ratios are prescribed as fixed values derived from empirical data and do not vary with environmental conditions. Although short-term environmental stress can transiently increase e.g., DOC exudation, such plastic responses are energetically costly and unlikely to persist over the multi-decadal timescales relevant to climate change. Evolutionary adaptation is expected to favor more efficient carbon allocation, supporting the use of fixed production ratios as a pragmatic approximation, which we note as an area for future refinement.

**Figure 4.** Present-day average of depth-integrated biological sources and sinks of DOC over the upper 100 m of the ocean, and their projected future changes. The figure of present-day averaged depth-integrated DOC production rates represents the combined contributions of biological DOC sources, including phytoplankton exudation, zooplankton egestion, and viral losses (a). Present-day average of depth-integrated DOC remineralization as the only biological sink (c). Right panel figures show projected future changes to DOC production (b) and remineralization rates (d), calculated as difference between future and present-day averages.

# 3.3 Linking biological constraints to physical DOC export

Isolating the trends of decreasing DOC production and a slower-decreasing remineralization (at approximately half the rate of production) reveals an overall decline in global net DOC production (calculated as the difference between production processes and remineralization; Fig. 5a). Yet, DOC concentration is simulated to increase in vast parts of the surface ocean (Fig. 5b). Ocean circulation plays a crucial role in redistributing and exporting DOC, significantly influencing its overall distribution, and concentration in the water column. When accounting for all fluxes affecting DOC, including advection- and mixing-mediated export (hereafter referred to as DOC export), and constant riverine input, upper 100 m DOC concentration shows an increasing trend of 0.04% per year (Fig. 5b).

DOC export through 100 m depth, on the other hand, is projected to decrease at a rate of -0.07% per year (Fig. 5c). The following section will present our findings on the distribution and dynamics of DOC export through 100 m and 1000 m depths, along with future projections.

**Figure 5.** Time series of DOC variables linking biological constraints to physical DOC export from 1990-2100. Net DOC production is calculated solely as a difference between biological sources and sink of DOC averaged over the upper 100 m of the ocean (a). DOC concentration is a net results of all the biogeochemical and physical processes that affect DOC pools. Presented is the DOC concentration averaged over the upper 100 m (b). DOC export shows the sum of export processes that transport DOC across 100 m depth (c). All time series are expressed as % change, where the baseline average was calculated as present-day average (1990-2010) based on which the relative changes were obtained. Trends were calculated using linear regression, with regression lines shown as dashed lines and the average trend values indicated in their corresponding panels. Note the different y-axes used in the panels for clarity.

#### 295 **3.4 DOC export**


The model simulates a present-day global DOC export of 1.6 PgC yr<sup>-1</sup> at 100 m depth (Fig. 6a). For comparison, total oceanic DOC inventory is 704 PgC in our model, illustrating the scale of annual export in relation to the total DOC reservoir and its role in the marine carbon cycle. Our DOC export rate estimate agrees closely with the value reported by DeVries and Weber (2017) and remains within the range of previous estimates but slightly lower than 1.9 PgC yr<sup>-1</sup> and 1.8 PgC yr<sup>-1</sup> reported by data-constrained global circulation models (Nowicki et al., 2022; Hansell et al., 2009, respectively), and  $2.3 \pm 0.6$  PgC yr<sup>-1</sup> estimated using an artificial neural network-constrained biogeochemistry model (Roshan and DeVries, 2017). Studies using ESMs (CESM2 (Sreeush et al., 2024), ESM4-COBALTv2 (Stock et al., 2020), and IPSL-PISCES-v2 (Aumont et al.,

2015)) reported DOC export rates of 2.0, 3.6, and 0.96 PgC yr<sup>-1</sup>, respectively. While these studies provide valuable estimates under present-day conditions, the responses of DOC export and the underlying mechanisms to climate change remain sparsely investigated using global models.






Our model projects a modest decline (-0.1 PgC yr<sup>-1</sup>) in the global 100 m DOC export when comparing present-day and future averages, resulting in a 1.5 PgC yr<sup>-1</sup> export rate (Fig. 6b; Fig A3), with mixing-mediated export showing greater sensitivity to future climate forcing than advective transport. To our knowledge, the only other study that examines future DOC export changes reports a decrease from 2.0 PgC yr<sup>-1</sup> to 1.9 PgC yr<sup>-1</sup> under the SSP3-7.0 scenario. Considering that both projections reflect a strong climate change signal in high-emission futures, the projected decreases of 7% and 5% support the expectation of a continued downward trend.

Although export is conventionally reported at 100 m depth, it is also useful to analyze export at 1000 m, as it provides insight into longer-term carbon sequestration through DOC. Our simulations estimate a present-day export of 0.09 PgC yr<sup>-1</sup> at 1000 m depth (Fig. 6c). Export through 1000 m depth is approximately 20 times lower than at 100 m, indicating that DOC is overall highly susceptible to rapid remineralization (Lefévre et al., 1996). This suggests that DOC export is not an efficient carbon sequestration pathway, as it generally does not reach great depths, except in deep-water formation regions (Hansell et al., 2009). Indeed, the export is highest in the North Atlantic, accounting for roughly 60% (0.048 PgC yr<sup>-1</sup>) of global export at 1000 m depth, followed by high rates in the Weddell gyre (Fig. 6c). In future projections, there is a 25% decrease in 1000 m export (from 0.09 to 0.06 PgC yr<sup>-1</sup>; Fig. 6d; Fig. A4).

Despite good agreement with present-day observations, applying temperature sensitivity only to labile DOC may limit the accuracy of future projections. Semi-labile and semi-refractory DOC pools have empirically-based constant decay rates implicitly reflecting contemporary environmental conditions, so future temperature-driven changes are only partly captured through remineralization of these pools into labile DOC. Although this approach remains valid for contemporary DOC cycling, the lack of dynamic or mechanistic process representation adds uncertainty to climate change projections. Addressing this limitation, ongoing development of our model aims to incorporate environmentally sensitive decay rates for semi-labile and semi-refractory DOC pools to enhance the robustness of future projections.

**Figure 6.** Present-day DOC export rates across the ocean at 100 m (a) and 1000 m depth (c), and their projected future changes (b and d, respectively). Negative export values indicate upward DOC transport through these depths. The changes (b, d) are calculated as the difference between future and present-day averages. We overlay dots to indicate regions where changes in DOC export are statistically significant ( $\alpha = 0.05$ ), given the patchy nature of the projected changes.

The relative contributions of physical processes driving DOC export vary regionally and seasonally, and are expected to shift under climate change. Here, we further decompose these physical processes driving DOC export as advection and mixing fluxes. At 100 m depth, present-day global advective flux is 0.55 PgC yr<sup>-1</sup> (Fig. 7a), while the mixing flux is nearly double at 1.02 PgC yr<sup>-1</sup> (Fig. 7e). At 1000 m depth, in contrast, DOC export is dominated by advection (0.06 PgC yr<sup>-1</sup>; Fig. 7c) rather than mixing (0.02 PgC yr<sup>-1</sup>; Fig. 7g). This shift reflects the decreasing influence of vertical mixing with depth and the growing importance of large-scale circulation features such as the Atlantic Meridional Overturning Circulation (AMOC).



In the future scenario, the DOC advective flux at 100 m remains stable at  $0.56 \text{ PgC yr}^{-1}$  (Fig. 7b), whereas the DOC mixing flux decreases to  $0.94 \text{ PgC yr}^{-1}$  (Fig. 7f). At 1000 m, mixing remains nearly unchanged (Fig. 7h), while advection drops from  $0.06 \text{ to } 0.04 \text{ PgC yr}^{-1}$  (Fig. 7d).

The differential response of advection- and mixing-mediated DOC export to climate change can be tied to several mechanisms. Enhanced surface warming increases upper ocean stratification (Fig. A2), which inhibits vertical mixing and reduces DOC transport from the surface to the deeper layers. At the same time, projected weakening of the AMOC under warming scenarios decreases the advective transport at depth (Caesar et al., 2021; Rahmstorf et al., 2015). This is evident in the North Atlantic, where reduced formation of North Atlantic Deep Water diminishes an important pathway for deep DOC export (Fig. 7d). Furthermore, changes in wind patterns affect the intensity of upwelling in regions such as the Southern Ocean (Morrison and Hogg, 2013), further affecting advective fluxes.

To better understand the spatial expression of these processes, we next examine regional DOC export patterns. At 100 m, DOC export varies regionally due to a combination of physical transport and biological production dynamics. The highest export rates (10–20 gC m<sup>-2</sup> yr<sup>-1</sup>) occur in subtropical gyres (Fig. 6a). DOC export in these regions is primarily advective, consistent with large-scale circulation and wind-driven convergence zones. Advection also plays a role in transport in regions with strong ocean currents, for example, along the southern flanks of the Gulf Stream and Kuroshio. In these areas, DOC is transported laterally and vertically along isopycnal surfaces, contributing to local subduction and export at 100 m depth. In western boundary currents such as the Gulf Stream and Kuroshio, and along the path of the Antarctic Circumpolar Current (ACC), our simulations also show hotspots of mixing-driven export (15-25 gC m<sup>-2</sup> yr<sup>-1</sup>), likely due to strong eddy fields and intense vertical exchange, although eddy-driven processes are not explicitly resolved at our model resolution and are instead parameterized. Based on our model, mixing-driven export exceeds advection-driven export, underscoring the role of turbulent mixing in transferring biologically produced DOC below 100 m. This is consistent with strong vertical DOC gradients at the base of the mixed layer, especially in regions of seasonal deepening, where winter entrainment redistributes DOC accumulated during growing season. Winter deepening has also been shown to contribute to organic carbon transport in subpolar gyres and parts of the Southern Ocean via entrainment of surface accumulated DOC (Bif and Hansell, 2019; Levy et al., 2013).

However, our model's nominal 1° resolution potentially underestimates the contribution of mixing, as it does not resolve submesoscale eddies and associated vertical velocities that enhance local mixing and transport (Resplandy et al., 2019). These processes are especially important in frontal zones and regions of strong density gradients (Lévy et al., 2012). Despite this limitation, the projected trends support the view that changes in large-scale circulation will dominate future shifts in DOC export at 1000 m depth rather than local mixing processes.

**Figure 7.** Global patterns of advection- and mixing-mediated DOC export processes and their projected changes across two depth horizons. (a-d) DOC export via advection: present-day rates at 100 m (a) and 1000 m depth (c), and their projected future changes (b,d respectively). (e-h) DOC export via mixing: present-day rates at 100 m (e) and 1000 m depth (g), and their projected future changes (f,h respectively). All rates are in gC m<sup>-2</sup> yr<sup>-1</sup>. Negative values indicate upward DOC transport via advection through the specified depths.

Our results indicate that DOC export at 100 m contributes 25 19% to TOC export globally based on the present-day average (Fig. 8a), which is slightly higher but consistent with previous estimates of approximately 20% (Roshan and DeVries, 2017; Hansell et al., 2009). The fraction of DOC contribution to TOC export is highest in subtropical gyres, reaching up to 40 more than 70%, reflecting the subduction of accumulated DOC transported from high productivity regions by poleward surface flows, but also very low POC export rates in this area. Roshan and DeVries (2017) report this contribution reaching up to 70% in the subtropical ocean in their study. To support this result, we also provide a map of POC export distribution at 100 m depth, together with a map of its future changes in Appendix (Fig. A5). Globally integrated POC export flux in our model is 8.6 Pg yr<sup>-1</sup>. Equatorial upwelling counteracts export in these regions by bringing DOC back to the surface, as indicated by the negative values in Fig. 6a. In figure 8b we show the spatial pattern of future regional shifts in the relative contribution of DOC to TOC export flux. However, the globally integrated average change in the ratio of DOC to TOC export is <1% decrease increase (Fig. 8b). We do not present maps of DOC contribution to export at 1000 m depth, as it is very low (< 0.01%). This further emphasizes that export of labile, semi-labile, or semi-refractory DOC is not an efficient long-term carbon sequestration pathway, as previously discussed and corroborated by other studies (Hansell et al., 2009; Nowicki et al., 2022). However, the model estimates that 2.2 PgC yr<sup>-1</sup> of DOC is remineralized within the 100-1000 m depth window, with 1.6 PgC yr<sup>-1</sup> exported from the surface and an additional 0.6 PgC yr<sup>-1</sup> produced locally. Therefore, the substantial quantity of exported DOC plays an important ecological role as a source of organic matter for mesopelagic organisms (Santana-Falcón et al., 2017; Carlson et al., 2010).


**Figure 8.** Global distribution of DOC export contribution to total organic carbon export at 100 m: present-day ratio (a) and its projected future change (b).

## 4 Conclusions





This study analyses the combined influence of biological and physical processes in controlling DOC cycling, both under present-day conditions and in a future climate scenario explores the roles of specific biological and physical processes in shaping. Results indicate that while biological production of DOC is highest in nutrient-rich equatorial upwelling zones and along active boundary currents, its surface and depth distribution, and long-term fate are largely determined by physical transport. The accumulation of DOC in stratified subtropical gyres, driven by downwelling and subduction, illustrates the important interaction between biogeochemical production and ocean circulation in sustaining carbon reservoirs in the upper ocean. The model reproduces the large-scale spatial patterns of observed DOC reasonably well. However, noticeable differences in high-latitude regions point to remaining uncertainties, especially regarding the role of riverine carbon inputs. These findings emphasize the need for improved representation of land-ocean carbon fluxes, particularly in Arctic regions where seasonal river discharge plays a major role in DOC variability. Further insight is gained from the analysis of DOC export processes. In the near-surface layers (top 100 m), DOC export is mainly influenced by seasonal mixing and vertical turbulent exchange. In contrast, export at greater depths is dominated by advective processes linked to deep-water formation and large-scale circulation. The projection under the RCP8.5 climate change scenario shows a general decline in DOC export, reflecting both the impact of reduced vertical mixing and circulation slowdown. Despite these changes and a decline in net biological DOC production, modest increases in surface DOC concentrations suggest that physical redistribution processes may moderate upper-ocean DOC levels to some extent. We recognize a limitation in our analyses concerning the dynamics of semi-labile and semi-refractory DOC pools, whose degradation rates are not explicitly sensitive to environmental drivers. Consequently, any conclusion regarding contribution of DOC to long-term carbon sequestration must be interpreted with caution. While our results suggest that rapid remineralization of labile DOC limits its contribution to long-term carbon storage, the role of the more refractory DOC pools, though not dynamically represented, may still be significant for carbon export. Anyhow, the role of DOC in upper-ocean carbon export remains ecologically and biogeochemically relevant. Overall, the findings support the view that DOC has a limited role in long-term carbon sequestration due to its relatively fast remineralization through depth, although the more refractory pools can contribute to export substantially. However, its role in the upper-ocean carbon export remains ecologically and biogeochemically relevant. Also, its importance and contribution to TOC export varies regionally, with highest accumulation and contribution in subtropical gyres (25 19% of TOC export due to DOC). Some limitations of this study stem from, for example, subgrid-scale processes such as submesoscale mixing that are not fully resolved, and the assumptions about biochemistry and lability of DOC pools and their decay remain simplified. Improving the representation of these processes, for example through higher-resolution modeling and updated parameterizations, or the inclusion of mechanistic formulations, will be beneficial to better understand the role of DOC in the marine carbon cycle under ongoing climate change.

# Appendix A: Additional figures

**Figure A1.** Comparison of present-day modeled nutrient and chlorophyll concentration with observational data. Model skill metrics including spatial correlation (r), root mean square error (rmse), and bias are calculated and shown for each of the nutrients evaluated and chlorophyll. Macronutrient data were obtained from World Ocean Atlas 2018 (Garcia et al., 2019). For chlorophyll data we used (½) merged satellite product with Case 1 algorithm from GlobColour. GlobColour data (http://globcolour.info) used in this study has been developed, validated, and distributed by ACRI-ST, France.

ESM2M has been evaluated extensively in Dunne et al. (2012). However, COBALTv2 has not been operated within ESM2M before, as it has been developed for use in the successor ESM4. We therefore provide an evaluation of the most important variables for this study, where we show that overall the model is in good agreement with observations. Besides DOC concentrations,

the variables considered were macronutrient (i.e., nitrate, phosphate and silicate) and chlorophyll concentrations, averaged over the upper 100 m. Skill metrics we consistently considered were Pearson's correlation coefficient for spatial patterns, root mean square error for the data spread, and bias for the 'direction' of model errors.

**Figure A2.** Projected future changes in sea-surface temperature (SST) in  $^{\circ}$  C (a) and mixed-layer depth in m (b).

**Figure A3.** Timeseries of regionally-integrated DOC export through 100 m depth expressed as percent change relative to 1990–2010 mean. Regions considered are North Pacific, North Atlantic, Subtropical, Equatorial, and Southern Ocean regions. Dashed lines indicate linear trends over the analyzed period.

**Figure A4.** Timeseries of regionally-integrated DOC export through 1000 m depth expressed as percent change relative to 1990–2010 mean. Regions considered are North Pacific, North Atlantic, Subtropical, Equatorial, and Southern Ocean regions. Dashed lines indicate linear trends over the analyzed period.

Figure A5. Present-day POC export flux at 100 m depth (a) and its projected future change (b).

**Table A1.** Model tuning: parameter values used across model configurations.

| Parameter                                                | ESM2M-COBALTv1                            | ESM4-COBALTv2                            | ESM2M-COBALTv2                              |
|----------------------------------------------------------|-------------------------------------------|------------------------------------------|---------------------------------------------|
| beta_fescav (Iron scavenging rate)                       | 0.0                                       | 2.5e9                                    | 0.80e9                                      |
| P_C_max_Di (Diazotroph max C prod. rate)                 | $0.50/\mathrm{sperd~sec^{-1}}$            | $0.70/\text{sperd sec}^{-1}$             | $0.50$ /sperd sec $^{-1}$                   |
| ca_2_n_arag (Ca:N ratio, aragonite)                      | $0.020\times106/16$                       | $0.050\times106/16$                      | $0.040\times106/16$                         |
| ca_2_n_calc (Ca:N ratio, calcite)                        | $0.010\times106/16$                       | $0.015\times106/16$                      | $0.014\times106/16$                         |
| gge_max_bact (Max. bacterial growth efficiency)          | 0.4                                       | 0.3                                      | 0.4                                         |
| bresp_bact (Bacterial respiration rate)                  | $0.0075/\mathrm{sperd~sec^{-1}}$          | $0.0$ /sperd sec $^{-1}$                 | $0.0075/\mathrm{sperd~sec^{-1}}$            |
| fe_coast (Coastal iron source)                           | $1 \times 10^{-11}$                       | 0                                        | $1 \times 10^{-11}$                         |
| vir_Sm (Small phyto. viral mortality)                    | $0.025\times10^6/\mathrm{sperd~sec^{-1}}$ | $0.10\times10^6/\mathrm{sperd~sec^{-1}}$ | $0.20 	imes 10^6 / \mathrm{sperd~sec^{-1}}$ |
| vir_Bact (Bacterial viral mortality)                     | $0.033\times10^6/\mathrm{sperd~sec^{-1}}$ | $0.10\times10^6/\mathrm{sperd~sec^{-1}}$ | $0.20 	imes 10^6 / \mathrm{sperd~sec^{-1}}$ |
| phi_det_smz (Small zoo. detrital prod.)                  | 0                                         | 0.05                                     | 0.2                                         |
| <pre>phi_det_mdz (Medium zoo. detrital prod.)</pre>      | 0.2                                       | 0.2                                      | 0.25                                        |
| <pre>smz_ipa_bact (Small zoo. bact. ingest. pref.)</pre> | 0.25                                      | 0.5                                      | 0.25                                        |
| phi_ldon_smz (Small zoo. LDON prod.)                     | 0.165                                     | 0.175                                    | 0.07                                        |
| <pre>phi_ldon_mdz (Medium zoo. LDON prod.)</pre>         | 0.055                                     | 0.07                                     | 0.035                                       |

- . LF and CL designed the study, with significant input from AW. AW configured the model set-up used for this study and ran the simulations. LF processed model output and conducted analyses of the study. LF wrote the initial manuscript draft. All authors discussed results and contributed to revising, editing, and writing of the paper.
- 420 . The authors have no competing interests to declare.

- . We acknowledge the outstanding efforts of the members of the Geophysical Fluid Dynamics Laboratory in developing ESM2M and making the code publicly available. We also highly appreciate Charlie Stock's invaluable assistance in setting up the model and adjusting the COBALTv2 parameterizations and input files for use within ESM2M. This work was supported by a grant from the Swiss National Supercomputing Centre (CSCS) under project ID sm85 on Alps and Piz Daint. This work was supported by the Swiss State Secretariat for Education, Research and Innovation (SERI) under contract number MB22.00069. C.L. acknowledges support from the Swiss National Science Foundation under grant 203448.
- . The dissolved organic carbon (DOC) compilation dataset obtained from global ocean observations are provided by Hansell et al. (2021). Macronutrient data were sourced from the World Ocean Atlas 2018 (Garcia et al., 2019). Chlorophyll-a data were derived from the GlobColour merged satellite product (1/4° resolution, Case 1 algorithm), developed and distributed by ACRI-ST, France. The COBALTv2-430 ESM2M model outputs used in the figures are publicly accessible via Zenodo: https://doi.org/10.5281/zenodo.15150328. All other model outputs generated during this study are available from the corresponding author upon request.

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
