# Peer review of "Dissolved organic carbon dynamics in a changing ocean: A COBALTv2 - ESM2M An ESM2M-COBALTv2 coupled model analysis"

_EGUsphere, 2025_

## Author Response (AR1)

**Response to the review #1**

The manuscript bei Flaniak et al. describes simulations of the dissolved organic carbon (DOC) in an Earth System Model. In particular, the study focuses on DOC export in the ocean under present and future conditions. The authors conclude the DOC export in the upper ocean is reduced in a future climate scenario due to intensified stratification and reduce nutrient supply, however, DOC concentrations slightly increase in the upper Ocean due to physical transport. The authors took great care to embed the results into a larger context and compare to existing literature. The study is carefully conceived, well written and the manuscript is clearly structured. It addresses a very timely and relevant question and fits well to the scope of the journal. I therefore recommend publication after some comments have been addressed:

**Main comments:**

 Methods: How is DOC export calculated? Please describe the approach in the methods. Opposite to POC export, DOC is also transported upwards again, is this taken into account (I assume yes, given the negative values, l. 328)?

We have clarified our methods to explicitly describe how upward as well as downward transport of DOC is accounted for, as follows (L 140-149):

"To quantify the physical transport mechanisms of DOC, we extracted the vertical advective and turbulent mixing fluxes directly from the MOM5 ocean model component using a Finite Volume Method. For advective fluxes, the volume flux across each cell face is multiplied by the DOC concentration of the donor cell, capturing both horizontal and vertical transport. Turbulent mixing fluxes are derived from the internal physical mixing model, which includes vertical diffusion, K-profile parameterisation, and additional subgrid-scale processes (Dunne et al., 2012). DOC export at specified depth horizons (100 m and 1000 m) is calculated by identifying the cell edge closest to the target depth and accumulating the exact flux through that level as a diagnostic at the model's output frequency. This method captures both downward and upward DOC transport; negative flux values indicate net upward movement (e.g., due to upwelling or turbulent mixing). The Finite Volume implementation ensures exact budget closure to machine precision, thereby enabling a precise separation of advective and mixing contributions to DOC export within each fixed-depth control volume."

• Inventory versus DOC export: when DOC is exported, a part of it is remineralised to DIC. The focus on export alone may bias the overall picture, as described in a recent study for POC export (Frenger, Landolfi et al., 2024). In order to assess the impact on the carbon cycle, it would be informative to assess the DOC inventory as a whole. I recommend to add these values for comparison.

We appreciate the point raised in the cited paper (Frenger et al., 2024) regarding the limitations of using only the (downward) export flux as a diagnostic for the biological carbon pump. In our study, however, the calculation of DOC export is based on the net

vertical flux across the specified depth horizons (in our case 100 m and 1000 m). This means that our approach integrates the total DOC flux, so any DOC that is transported downward and subsequently returns upward via ocean circulation or mixing is included in the calculation. Therefore, the net DOC export value we report represents the balance between downward export and upward return, rather than just the gross downward flux.

In response to your suggestion, we have included the global DOC inventory value from our model simulations (~ 704 PgC stored as DOC in the ocean) to provide a context for the relative magnitude of annual DOC export, as follows (L 287-288):

"For comparison, total oceanic DOC inventory is 704 PgC in our model, illustrating the scale of annual export in relation to the total DOC reservoir and its role in the marine carbon cycle."

- DOC dynamics: The model is based on a number of assumptions for the DOC pool for which evidence is lacking. This is not only a problem of this particular model, but is common in many DOC models, due to the diversity in composition and bacterial consumers that is challenging to adequately reflect in models. Still, these assumptions may affect the overall results, and I therefore recommend to clearly discuss which result depends on which assumptions, stating potential systemic uncertainties in the modeling approach. In particular:
  - why is only the labile pool temperature dependent? The authors reason that this is the only pool that depends on bacterial metabolism. However, the primary control of DOC degradation in the ocean is microbial consumption – what is the rationale of omitting the temperature dependency for the semi-labile and semi-refractory fraction, and how would a temperature dependency qualitatively affect the results? Lonborg et al. also project a temperature dependency of the refractory pool.

We acknowledge the reviewer's point regarding the lack of explicit temperature dependence for semi-labile and semi-refractory DOC remineralization in our current model. While the current decay rates for these pools in the model are based on empirical observations, they do not respond dynamically to environmental drivers. Addressing this, we are now developing a new model framework in which the remineralization rates of semi-labile and semi-refractory DOC will be made environmentally sensitive, allowing them to vary as a function of temperature and nutrient conditions. The parameters related to temperature-sensitive remineralization are adopted exactly from Lonborg et al. (2018). This ongoing work aims to better capture the complexity of DOC cycling under changing climate scenarios. We added sentences where we discuss the limitations of static semi-labile and semi-refractory pools, especially in context of future projections. This discussion was added to the end of chapter 3.2 (i.e. Biological sources and sinks as primary constraints on DOC concentration and distribution) (L262-269).

"These projected trends highlight the interplay between physical and biogeochemical drivers in shaping DOC dynamics. In interpreting these results, it is important to consider how certain model assumptions may influence the robustness of future projections.

Our modeled distribution of DOC and other relevant biogeochemical variables aligns well with present-day observations (Fig. 2; Appendix, Fig. A1), although temperature-sensitive remineralization is currently applied only to labile DOC. Semi-labile and semi-refractory DOC pools use empirically based constant decay rates reflecting current conditions, so future temperature effects are partially captured through remineralization into labile DOC. While this represents a suitable method for analysis of present-day cycling, it may introduce some uncertainty in future projections. Ongoing model development aims to incorporate environmentally sensitive decay rates for these pools to improve projection robustness."

The DOC production results from fixed, prescribed ratios of NPP or zooplankton egestion. As production and consumption rates are decoupled for the semi-labile and semi-refractory pools, any change in the production rate would alter the overall DOC concentration (other than the labile pool where the degradation rate is coupled due to the explicit calculation of the bacteria uptake). How certain is the assumption that these production ratios remain fixed in a future scenario?

We understand reviewer's concern centers on the certainty of assuming fixed, prescribed DOC production ratios (from NPP or zooplankton egestion) in future climate scenarios, also since production and consumption rates are decoupled for the semilabile and semi-refractory pools.

Our rationale for this is as follows:

While it is well-documented that DOC release can increase transiently under stress conditions such as nutrient limitation or temperature stress, these are short-term (plastic) responses. Such stress-induced DOC release is generally considered metabolically costly (a 'waste of energy') and not optimal over extended periods. Over the multi-decadal timescales relevant to climate change, evolutionary processes are expected to favor metabolic efficiency when adapting to changing conditions and thereby also minimize energetically wasteful responses like excessive DOC release. For example, phytoplankton have demonstrated rapid adaptive potential (i.e., within 100–1000 generations) which suggests that populations exposed to chronic environmental change would evolve towards optimized carbon allocation rather than persistently elevated DOC exudation (e.g., Schaum et al., 2018). Given this context, we consider fixed DOC production ratios to be a reasonable approximation for projecting future ocean conditions over multi-decadal periods. However, we recognize the need and value of future work to explore environmentally sensitive production ratios and their potential impacts on DOC cycling.

We have clarified this point and added it as an area for future refinement, at the end of chapter 3.2 (L269-274).

"Similarly, DOC production ratios are prescribed as fixed values derived from empirical data and do not vary with environmental conditions. Although short-term environmental stress can transiently increase e.g., DOC exudation, such plastic responses are energetically costly and unlikely to persist over the multi-decadal timescales relevant to climate change. Evolutionary adaptation is expected to favor more efficient carbon allocation, supporting the use of fixed production ratios as a pragmatic approximation, which we note as an area for future refinement."

• Conclusion on the role of DOC in carbon sequestration, especially in l. 352: The conclusion that the DOC does not contribute much to carbon sequestration is a direct consequence of how the model is set up. The processes resolving the actual sequestration of the carbon pool that make up most of the inventory, i.e. the dynamics of the semi-refractory and refractory carbon stock, are not fully modelled explicitly, and therefore do only dynamically respond to changes in production, not in degradation. Therefore, a conclusion on the overall contribution of DOC to carbon sequestration is not possible by this model approach. I therefore recommend to scale back this conclusion, as it strictly only applies to the labile fraction, which is not expected to contribute much to the storage due to its small inventory size.

We agree with the reviewer's comment on the extent of our conclusion regarding carbon sequestration considering the model limitations and we accordingly adjust the conclusion by modifying/adding these explanations (L381-385):

"We recognize a limitation in our analyses concerning the dynamics of semi-labile and semi-refractory DOC pools, whose degradation rates are not explicitly sensitive to environmental drivers. Consequently, any conclusion regarding contribution of DOC to long-term carbon sequestration must be interpreted with caution. While our results suggest that rapid remineralization of labile DOC limits its contribution to long-term carbon storage, the role of the more refractory DOC pools, though not dynamically represented, may still be significant for carbon export."

Specific comments:

l. 91 a word is missing in this sentence

We have added a missing word (L 91).

l. 99: I might misunderstand sth here, but isn't a higher half saturation constant a disadvantage (i.e., not a benefit), because it takes higher substrate concentrations to reach the maximum uptake rate? Should it actually be "lower half-saturation constants" showing a higher substrate specificity, or is it indeed higher, following the trade-off of

high maximum growth rate and low affinity (=high half-saturation constant). Please doublecheck and correct the sentence or the reasoning if required.

We thank the reviewer for catching this mistake. It is ineed a 'disadvantage' to have a higher half saturation constant and large phytoplankton have 5 times higher half-saturation constant than small phytoplankton, so there was a mistake in constructing this sentence.

It is corrected as follows (L 99):

"The uptake of nutrients is modeled after Michalis-Menten kinetics, assigning significantly lower half-saturation constants to small phytoplankton to represent the benefits of higher surface area to volume ratio in nutrient uptake."

l. 176: The Lennartz at al study discussed the counteracting effects of temperature dependences on growth rate and growth efficiency, which was then not considered in the main model, which is only using macronutrient colimitation – please adjust the decription. Their overall correlation coefficient was R2=0.75, the R2=0.55 was for the surface ocean - please doublecheck which one to compare to in your case.

We corrected the sentence referring to performance of the modified ESM used in Lennartz et al. (2024) study as follows (L 183-185):

"Lennartz et al. (2024) used a modified ESM incorporating macronutrient co-limitation on DOC uptake, thereby achieving a spatial correlation of R2 = 0.55 for the surface ocean and R2 = 0.75 when integrated over depth."

l. 193: Is there a word missing, i.e. "The subtropical ocean..." or "subtropical oceanS"?

The beginning of this sentence was corrected as "The subtropical ocean(...)".

l. 215: The text refers to figure 3c which is not present in Fig. 3, I assume Fig. 4 is meant. Also: here you describe DOC uptake is co-limited by nutrient availability, is this part of your model.

We thank the reviewer for catching this; the reference was corrected to Fig. 4, and yes, the co-limited DOC uptake is part of our model.

l. 219, l. 239: please doublecheck figure numbering, I assume Fig. 4 is meant.

These were also corrected to refer to Fig. 4.

l. 338: I recommend to reformulate the first sentence. Only the labile component actually resolves both biological production and consumption processes, whereas the other pools that make up most of the standing stock decay with first-order degradation rates not resolving degradation processes processes.

We rewrote the first sentence of the conclusion more carefully (L367-368). This wording intentionally reflects that only certain biological processes are explicitly resolved, while others (i.e., degradation of more recalcitrant pools) are treated more simply. We think, as an opening sentence to the conclusion this formulation is careful enough, considering that those specific limitations are addressed earlier in the discussion as well as later in conclusions.

"This study explores the roles of specific biological and physical processes in shaping DOC cycling, both under present-day conditions and in a future climate scenario."

Figure 3: I recommend to flip the axis to make the reference DOC the independent variable.

The figure was modified according to the reviewer's recommendation.

Schaum, C. E., Buckling, A., Smirnoff, N., Studholme, D. J., & Yvon-Durocher, G. (2018). Environmental fluctuations accelerate molecular evolution of thermal tolerance in a marine diatom. Nature Communications, 9(1), 1719. DOI: 10.1038/s41467-018-03906-5

**Response to the review #2**

First, "thank you" to the authors and editor for patiently waiting for my comments. My apologies for the slight delay in posting this.

Flanjak et al. investigates the spatial distribution of dissolved organic carbon in the global ocean, its export at 100 and 1000m, and their potential future changes using a fully coupled Earth System Model, namely ESM2M-COBALTv2, and discusses the physical and biogeochemical drivers for such variability and changes. Their results show that simulated DOC export to the deep ocean (across 1000m) is quite small

compared to that across 100m, suggesting high remineralization of DOC in the upper ocean. Also, DOC export to the deep ocean is overall insensitive to global changes under RCP8.5 scenario. The authors conclude that bioactive DOC plays a limited role in sequestrating carbon into the deep ocean over long-term time scales. The manuscript is very well written, methods and results are easy to follow, and figures overall in good quality despite those listed in the minor comments. I recommend publication in EGUsphere after addressing a few minor issues.

**Main comment:**

I have one main comment regarding Figure 8.

In Line 324, the authors state that "our results indicate that DOC export at 100m contributes 25% to TOC export globally based on the present-day average (Fig. 8a)". I am not able to correspond this conclusion to what is shown in Figure 8a. Based on the colorbar, a majority of the ocean (>90%) shows a (DOC export/TOC export) value of 0-8% with some high values (8-30%) along the edges of the subtropical gyres. I do not think the global mean would equal 25%. It is more like a few percentage in my eyes. Is this a calculation error? Or is Figure 8a mistakenly showing results for 1000m? If former, this brings the DOC contribution to export down to almost none and revises a key conclusion of this study. Please check and elaborate on how the global mean is calculated. Also, I highly recommend that the authors use white in the center of the colorbar, or increase the resolution of the colorbar. With the one used in Figure 8, it is impossible to guess the value for the majority of the ocean and [0-8%] is quite a range to be presented as one color pixel.

We thank the reviewer for this careful observation, which helped us identify an error in our calculation of the DOC contribution to total organic carbon (TOC) export. In the final revised analysis, we now calculate the ratio as DOC export at 100 m divided by the sum of absolute values of DOC and POC export at 100 m, where POC is defined as the flux through the 100 m depth horizon. Therefore, we use the formulation:

DOC/TOC export ratio = DOC\_export,100m / (|DOC\_export,100m| + |POC\_export,100m| +  $\epsilon$ ), with  $\epsilon$ =10-12.

With this corrected calculation, the spatial distribution shows substantially higher DOC contributions to TOC export in the subtropical gyres, where DOC export has highest rates and POC export lowest. The revised global mean value is therefore consistent with the updated Figure 8a. To support this interpretation, we have added POC export at 100 m and its future changes as supplementary figures. We also made the colorbar resolution higher in Figure 8, to make the full range of values easier to interpret. We modified the corresponding part of the discussion accordingly.

**Minor comments:**

Title:

I think ESM2M-COBALTv2 is more appropriate than COBALTv2-ESM2M.

We accept reviewer's suggestion.

**Introduction:**

Line 35: "this process (enhanced primary production) is often accompanied by a larger proportion of organic matter being released in dissolved form rather than as particulate organic matter."

I don't seem to be able to follow why enhanced PP (under warming and increased CO2 level) increases the ratio of (DOC production/POC production) without reading the cited references. Is this a direct PP effect, or rather a warming/CO2 effect? Could use some brief explanations here.

We intended to highligh that, almost by default, production of DOC increases as PP increases, but that the ratio of DOC/POC is affected as well. However, we agree that the latter point might be difficult to follow without reading the references, and it is also beyond the scope of this study. Since we do not revisit this point later in the manuscript, we have removed the sentence in question to avoid confusion: This process is often accompanied by a larger proportion of organic matter being released in dissolved form rather than as particulate organic matter.

**Methods:**

Line 89, Session 2.2.1 Ecosystem dynamics, could be streamlined a bit better given that COBALTv2 is well documented and cited, and what is presented in this session is not part of this work. Instead, I recommend that the authors expand session 2.2.3 on model tuning and add a session in the Results or supplementary materials to show which parameters are changed and how.

We appreciate the reviewer's suggestion regarding the streamlining of the ecosystem model description. However, we believe that retaining this section is important, as the

way that the ecosystem (phytoplankton, zooplankton, bacteria) is modeled directly and indirectly influences DOC production. By modifying the last sentence in this subchapter we try to clarify that development of the model is not a part of this work (L116-117). We agree with the reviewer's recommendation to enhance clarity regarding model tuning. To address this, we added a table in the Appendix summarizing the key parameters that were adjusted during the tuning process. This table provides a concise overview of which parameters were changed and their values across model versions.

"The development of the model was carried out and described in detail, together with additional ecosystem equations, by Stock et al. (2020)."

| Parameter                            | ESM2M-               | ESM4-                | ESM2M-               |
|--------------------------------------|----------------------|----------------------|----------------------|
|                                      | COBALTv1             | COBALTv2             | COBALTv2             |
| beta_fescav (Iron scavenging         | 0.0/spery            | 2.5e9/spery          | 0.80e9/spery         |
| rate)                                | sec -1    | sec -1    | sec -1    |
| P_C_max_Di (Diazotroph max C         | 0.50/sperd           | 0.70/sperd           | 0.50/sperd           |
| prod. rate)                          | sec -1    | sec -1    | sec -1    |
| ca_2_n_arag (Ca:N ratio,             | 0.020 × 106/16       | 0.050 ×              | 0.040 ×              |
| aragonite)                           |                      | 106/16               | 106/16               |
| ca_2_n_calc (Ca:N ratio, calcite)    | 0.010 × 106/16       | 0.015 ×              | 0.014 ×              |
|                                      |                      | 106/16               | 106/16               |
| gge_max_bact (Max. bacterial         | 0.4                  | 0.3                  | 0.4                  |
| growth efficiency)                   |                      |                      |                      |
| bresp_bact (Bacterial respiration    | 0.0075/sperd         | 0.0/sperd            | 0.0075/sperd         |
| rate)                                | sec -1    | sec -1    | sec -1    |
| fe_coast (Coastal iron source)       | 1E-11                | 0                    | 1E-11                |
| vir_Sm (Small phyto. viral           | 0.025×1e6/spe        | 0.10×1e6/spe         | 0.20×1e6/spe         |
| mortality)                           | rd sec -1 | rd sec -1 | rd sec -1 |
| vir_Bact (Bacterial viral mortality) | 0.033×1e6/spe        | 0.10×1e6/spe         | 0.20×1e6/spe         |
|                                      | rd sec -1 | rd sec -1 | rd sec -1 |
| phi_det_smz (Small zoo. detrital     | 0                    | 0.05                 | 0.2                  |
| prod.)                               |                      |                      |                      |
| phi_det_mdz (Medium zoo.             | 0.2                  | 0.2                  | 0.25                 |
| detrital prod.)                      |                      |                      |                      |
| smz_ipa_bact (Small zoo. bact.       | 0.25                 | 0.5                  | 0.25                 |
| ingestion pref.)                     |                      |                      |                      |
| phi_ldon_smz (Small zoo. LDON        | 0.165                | 0.175                | 0.07                 |
| prod.)                               |                      |                      |                      |
| phi_ldon_mdz (Medium zoo.            | 0.055                | 0.07                 | 0.035                |
| LDON prod.)                          |                      |                      |                      |

Line 138: Note that organic carbon (presented in nitrogen currency) from rivers in COBALTv2 as published in Stock et al. 2020 includes the labile, semi-labile, and semi-refractory components. Is this the case in this work?

That is the case here as well. We modified the sentence where we describe riverine input to include this information.

**Results:**

Line 185: need to clarify here that riverine DOC inputs to the ocean still resolve seasonality from dynamically changing freshwater flow calculated in the coupled model. The authors stated correctly that river DOC contributions do not have their seasonality, but could make it a bit more clear by saying something like "the model uses prescribed, climatological concentrations for river carbonate constituents, thus only resolves the temporal variability of river DOC contributions due to freshwater variability but not due to DOC concentrations".

We thank the reviewer for the elaborated suggestion to improve description of riverine input dynamics, and we modified this part accordingly (L 192-195).

"In contrast, the model (ESM2M-COBALTv2) does not dynamically resolve the riverine DOC contributions and their seasonality, but model uses prescribed, climatological concentrations for river carbonate constituents, thus only resolves the temporal variability of river DOC contributions due to freshwater variability but not due to DOC concentrations."

---

## Author Response (AR2)

Response to the reviewer and editor

We thank the reviewer for their constructive feedback and for recognizing our revisions. Below, we address the remaining two minor comments.

I would like to thank the authors for carefully addressing the comments in detail. I agree to the changes made to the manuscript.

Two minor points remain and I apologize for not being fully clear in my initial review:

• DOC inventory: Thanks for providing the total DOC inventory. My recommendation was rather to assess the DOC inventory in your future simulation, i.e. how much more or less carbon is stored, instead of only focusing on the DOC export. I recommend to add this information to the main text.

We have added this information to the main text as follows (L290-292):

"For comparison, the modeled total oceanic DOC inventory is 704 PgC under presentday conditions and decreases by 0.6 PgC in the simulated future period, illustrating the scale of annual export in relation to the total DOC reservoir and its role in the marine carbon cycle."

• Fixed production ratios of the non-bacterially mediated pools (i.e., the percentage of total production entering respective pools): The initial concern was not about evolutionary changes, but rather that these pools are modelled based on first-order kinetics, which is a concept derived from modelling \_net\_ removal rates (Hansell et al., 2012; Kirchman et al., 1993) that by definition obscures the underlying production and consumption (i.e. fixed parameters based on net removal dynamics). As your results show for the labile pool, production and consumption may respond differently to environmental drivers. As these pools form the largest part of the total DOC pool, their dynamics play a relatively large role in the carbon export you calculate. Extrapolating net removal rates into the future, without accounting for the distinct dynamics of these underlying processes, introduces uncertainty and should be acknowledged.

We agree with the reviewer. This uncertainty is already acknowledged in several parts of the discussion, and we have refined the relevant sections to emphasize the point more clearly.

L314-320: "Despite good agreement with present-day observations, applying temperature sensitivity only to labile DOC may limit the accuracy of future projections. Semi-labile and semi-refractory DOC pools have empirically based constant decay rates implicitly reflecting contemporary environmental conditions, so future temperature-driven changes are only partly captured through remineralization of these pools into labile DOC. Although this approach remains valid for contemporary DOC cycling, the lack of dynamic or mechanistic process representation adds uncertainty to climate change projections. Addressing this limitation, ongoing development of our model aims to incorporate environmentally sensitive decay rates for semi-labile and

semi-refractory DOC pools to enhance the robustness of future projections."

We also modified the following paragraph to further address the reviewer's concern (L264-270):

"In interpreting these results, it is important to consider how certain model assumptions may influence the robustness of future projections. Our modeled distribution of DOC and other relevant biogeochemical variables aligns well with present-day observations (Fig. 2; Appendix, Fig. A1), although temperature-sensitive remineralization is currently applied only to labile DOC. Semi-labile and semi-refractory DOC pools use empirically based constant decay rates reflecting current conditions, so future temperature effects are partially captured through remineralization into labile DOC. These decay rates follow first-order kinetics representing net removal, thereby implicitly combining production and consumption processes. While suitable for analysis of present-day cycling, this approach formulation may introduce uncertainty in future projections."

We finally refer to the specific uncertainty again in the conclusion (L386-389):

"We recognize a limitation in our analyses concerning the dynamics of semi-labile and semi-refractory DOC pools, whose degradation rates are not explicitly sensitive to environmental drivers. Consequently, any conclusion regarding contribution of DOC to long-term carbon sequestration must be interpreted with caution."